# CORRECTING EXPERIENCE REPLAY
# FOR MULTI-AGENT COMMUNICATION

**Sanjeevan Ahilan**
Gatsby Computational Neuroscience Unit
University College London
ahilan@gatsby.ucl.ac.uk

**Peter Dayan**
Max Planck Institute for Biological Cybernetics
and University of Tübingen
dayan@tue.mpg.de

## ABSTRACT

We consider the problem of learning to communicate using multi-agent reinforcement learning (MARL). A common approach is to learn off-policy, using data sampled from a replay buffer. However, messages received in the past may not accurately reflect the current communication policy of each agent, and this complicates learning. We therefore introduce a 'communication correction' which accounts for the non-stationarity of observed communication induced by multi-agent learning. It works by relabelling the received message to make it likely under the communicator's current policy, and thus be a better reflection of the receiver's current environment. To account for cases in which agents are both senders and receivers, we introduce an ordered relabelling scheme. Our correction is computationally efficient and can be integrated with a range of off-policy algorithms. We find in our experiments that it substantially improves the ability of communicating MARL systems to learn across a variety of cooperative and competitive tasks.

## 1 INTRODUCTION

Since the introduction of deep Q-learning (Mnih et al., 2013), it has become very common to use previous online experience, for instance stored in a replay buffer, to train agents in an offline manner. An obvious difficulty with doing this is that the information concerned may be out of date, leading the agent woefully astray in cases where the environment of an agent changes over time. One obvious strategy is to discard old experiences. However, this is wasteful – it requires many more samples from the environment before adequate policies can be learned, and may prevent agents from leveraging past experience sufficiently to act in complex environments. Here, we consider an alternative, Orwellian possibility, of using present information to correct the past, showing that it can greatly improve an agent's ability to learn.

We explore a paradigm case involving multiple agents that must learn to communicate to optimise their own or task-related objectives. As with deep Q-learning, modern model-free approaches often seek to learn this communication off-policy, using experience stored in a replay buffer (Foerster et al., 2016; 2017; Lowe et al., 2017; Peng et al., 2017). However, multi-agent reinforcement learning (MARL) can be particularly challenging as the underlying game-theoretic structure is well known to lead to non-stationarity, with past experience becoming obsolete as agents come progressively to use different communication codes. It is this that our correction addresses.

Altering previously communicated messages is particularly convenient for our purposes as it has no direct effect on the actual state of the environment (Lowe et al., 2019), but a quantifiable effect on the observed message, which constitutes the receiver's 'social environment'. We can therefore determine what the received message *would be* under the communicator's current policy, rather than what it was when the experience was first generated. Once this is determined, we can simply relabel the past experience to better reflect the agent's current social environment, a form of off-environment correction (Ciosek & Whiteson, 2017).

We apply our 'communication correction' using the framework of centralised training with decentralised control (Lowe et al., 2017; Foerster et al., 2018), in which extra information – in this case the policies and observations of other agents – is used during training to learn decentralised multi-

agent policies. We show how it can be combined with existing off-policy algorithms, with little computational cost, to achieve strong performance in both the cooperative and competitive cases.

## 2 BACKGROUND

**Markov Games** A partially observable Markov game (POMG) (Littman, 1994; Hu et al., 1998) for $N$ agents is defined by a set of states $\mathcal{S}$, sets of actions $\mathcal{A}_1, ..., \mathcal{A}_N$ and observations $\mathcal{O}_1, ..., \mathcal{O}_N$ for each agent. In general, the stochastic policy of agent $i$ may depend on the set of action-observation histories $H_i \equiv (\mathcal{O}_i \times \mathcal{A}_i)^*$ such that $\pi_i : \mathcal{H}_i \times \mathcal{A}_i \rightarrow [0, 1]$. In this work we restrict ourselves to history-independent stochastic policies $\pi_i : \mathcal{O}_i \times \mathcal{A}_i \rightarrow [0, 1]$. The next state is generated according to the state transition function $\mathcal{P} : \mathcal{S} \times \mathcal{A}_1 \times ... \times \mathcal{A}_n \times \mathcal{S} \rightarrow [0, 1]$. Each agent $i$ obtains deterministic rewards defined as $r_i : \mathcal{S} \times \mathcal{A}_1 \times ... \times \mathcal{A}_n \rightarrow \mathbb{R}$ and receives a deterministic private observation $o_i : \mathcal{S} \rightarrow \mathcal{O}_i$. There is an initial state distribution $\rho_0 : \mathcal{S} \rightarrow [0, 1]$ and each agent $i$ aims to maximise its own discounted sum of future rewards $\mathbb{E}_{s \sim \rho_{\boldsymbol{\pi}}, a \sim \boldsymbol{\pi}}[\sum_{t=0}^{\infty} \gamma^t r_i(s, \boldsymbol{a})]$ where $\boldsymbol{\pi} = \{\pi_1, ..., \pi_n\}$ is the set of policies for all agents, $\boldsymbol{a} = (a_1, ..., a_N)$ is the joint action and $\rho_{\boldsymbol{\pi}}$ is the discounted state distribution induced by these policies starting from $\rho_0$.

**Experience Replay** As an agent continually interacts with its environment it receives experiences $(o_t, a_t, r_{t+1}, o_{t+1})$ at each time step. However, rather than using those experiences immediately for learning, it is possible to store such experience in a replay buffer, $\mathcal{D}$, and sample them at a later point in time for learning (Mnih et al., 2013). This breaks the correlation between samples, reducing the variance of updates and the potential to overfit to recent experience. In the single-agent case, prioritising samples from the replay buffer according to the temporal-difference error has been shown to be effective (Schaul et al., 2015). In the multi-agent case, Foerster et al. (2017) showed that issues of non-stationarity could be partially alleviated for independent Q-learners by importance sampling and use of a low-dimensional 'fingerprint' such as the training iteration number.

**MADDPG** Our method can be combined with a variety of algorithms, but we commonly employ it with multi-agent deep deterministic policy gradients (MADDPG) (Lowe et al., 2017), which we describe here. MADDPG is an algorithm for centralised training and decentralised control of multi-agent systems (Lowe et al., 2017; Foerster et al., 2018), in which extra information is used to train each agent's critic in simulation, whilst keeping policies decentralised such that they can be deployed outside of simulation. It uses deterministic policies, as in DDPG (Lillicrap et al., 2015), which condition only on each agent's local observations and actions. MADDPG handles the non-stationarity associated with the simultaneous adaptation of all the agents by introducing a separate centralised critic $Q_i^{\boldsymbol{\mu}}(\boldsymbol{o}, \boldsymbol{a})$ for each agent where $\boldsymbol{\mu}$ corresponds to the set of deterministic policies $\mu_i : \mathcal{O} \rightarrow \mathcal{A}$ of all agents. Here we have denoted the vector of joint observations for all agents as $\boldsymbol{o}$.

The multi-agent policy gradient for policy parameters $\theta$ of agent $i$ is:

$$\nabla_{\theta_i} J(\theta_i) = \mathbb{E}_{\boldsymbol{o}, \boldsymbol{a} \sim \mathcal{D}}[\nabla_{\theta_i} \mu_i(o_i) \nabla_{a_i} Q_i^{\boldsymbol{\mu}}(\boldsymbol{o}, \boldsymbol{a})|_{a_i = \mu_i(o_i)}]. \tag{1}$$

where $\mathcal{D}$ is the experience replay buffer which contains the tuples $(\boldsymbol{o}, \boldsymbol{a}, \boldsymbol{r}, \boldsymbol{o'})$. Like DDPG, each $Q_i^{\boldsymbol{\mu}}$ is approximated by a critic $Q_i^w$ which is updated to minimise the error with the target.

$$\mathcal{L}(w_i) = \mathbb{E}_{\boldsymbol{o}, \boldsymbol{a}, \boldsymbol{r}, \boldsymbol{o'} \sim \mathcal{D}}[(Q_i^w(\boldsymbol{o}, \boldsymbol{a}) - y)^2] \tag{2}$$

where $y = r_i + \gamma Q_i^w(\boldsymbol{o'}, \boldsymbol{a'})$ is evaluated for the next state and action, as stored in the replay buffer. We use this algorithm with some additional changes (see Appendix A.3 for details).

**Communication** One way to classify communication is whether it is explicit or implicit. Implicit communication involves transmitting information by changing the shared environment (e.g. scattering breadcrumbs). By contrast, explicit communication can be modelled as being separate from the environment, only affecting the observations of other agents. In this work, we focus on explicit communication with the expectation that dedicated communication channels will be frequently integrated into artificial multi-agent systems such as driverless cars.

Although explicit communication does not formally alter the environmental state, it does change the observations of the receiving agents, a change to what we call its 'social environment'[1]. For agents which act in the environment and communicate simultaneously, the set of actions for each agent $\mathcal{A}_i = \mathcal{A}_i^e \times \mathcal{A}_i^m$ is the Cartesian product of the sets of regular environment actions $\mathcal{A}_i^e$ and explicit communication actions $\mathcal{A}_i^m$. Similarly, the set of observations for each receiving agent $\mathcal{O}_i = \mathcal{O}_i^e \times \mathcal{O}_i^m$ is the Cartesian product of the sets of regular environmental observations $\mathcal{O}_i^e$ and explicit communication observations $\mathcal{O}_i^m$. Communication may be targeted to specific agents or broadcast to all agents and may be costly or free. The zero cost formulation is commonly used and is known as 'cheap talk' in the game theory community (Farrell & Rabin, 1996).

In many multi-agent simulators the explicit communication action is related to the observed communication in a simple way, for example being transmitted to the targeted agent with or without noise on the next time step. Similarly, real world systems may transmit communication in a well understood way, such that the observed message can be accurately predicted given the sent message (particularly if error-correction is used). By contrast, the effect of environment actions is generally difficult to predict, as the shared environment state will typically exhibit more complex dependencies.

## 3 METHODS

Our general starting point is to consider how explicit communication actions and observed messages might be relabelled using an explicit communication model. This model often takes a simple form, such as depending only on what was communicated on the previous timestep. The observed messages $\boldsymbol{o}_{t+1}^m$ given communication actions $\boldsymbol{a}_t^m$ are therefore sampled (denoted by $\sim$) from:

$$\boldsymbol{o}_{t+1}^m \sim p(\boldsymbol{o}_{t+1}^m \mid \boldsymbol{a}_t^m) \tag{3}$$

Examples of such a communication model could be an agent $i$ receiving a noiseless message from a single agent $j$ such that $o_{i,t+1}^m = a_{j,t}^m$, or receiving the message corrupted by Gaussian noise $o_{i,t+1}^m \sim \mathcal{N}(a_{j,t}^m, \sigma)$ where $\sigma$ is a variance parameter. We consider the noise-free case in the multi-agent simulator for all bar one of our experiments, although the general idea can be applied to more complex, noisy communication models.

A communication model such as this allows us to correct past actions and observations in a consistent way. To understand how this is possible, we consider a sample from a multi-agent replay buffer which is used for off-policy learning. In general, the multi-agent system at current time $t'$ receives observations $\boldsymbol{o}_{t'}$, collectively takes actions $\boldsymbol{a}_{t'}$ using the decentralised policies $\boldsymbol{\pi}$, receives rewards $\boldsymbol{r}_{t'+1}$ (split into environmental rewards $\boldsymbol{r}_{t'+1}^e$ and messaging costs $\boldsymbol{r}_{t'+1}^m$) and the next observations $\boldsymbol{o}_{t'+1}$. These experiences are stored as a tuple in the replay buffer for later use to update the multi-agent critic(s) and policies. For communicating agents, we can describe a sample from the replay buffer $\mathcal{D}$ as the tuple:

$$(\boldsymbol{o}_t^e, \boldsymbol{o}_t^m, \boldsymbol{a}_t^e, \boldsymbol{a}_t^m, \boldsymbol{r}_{t+1}^e, \boldsymbol{r}_{t+1}^m, \boldsymbol{o}_{t+1}^e, \boldsymbol{o}_{t+1}^m) \sim \mathcal{D} \tag{4}$$

where we separately denote environmental ($e$) and communication ($m$) terms, and $t$ indexes a time in the past (rather than the current time $t'$). For convenience we can ignore the environmental tuple of observations, actions and reward as we do not alter these, and consider only the communication tuple $(\boldsymbol{o}_t^m, \boldsymbol{a}_t^m, \boldsymbol{r}_{t+1}^m, \boldsymbol{o}_{t+1}^m)$. Using the communication model at time $t'$, we can relate a change in $\boldsymbol{a}_t^m$ to a change in $\boldsymbol{o}_{t+1}^m$. If we also keep track of $\boldsymbol{a}_{t-1}^m$ we can similarly change $\boldsymbol{o}_t^m$. In our experiments we assume for simplicity that communication is costless (the 'cheap talk' setting), which means that $\boldsymbol{r}_{t+1}^m = 0$, however in general we could also relabel rewards using a model of communication cost $p(\boldsymbol{r}_{t+1}^m \mid \boldsymbol{a}_t^m)$. Equipped with an ability to rewrite history, we next consider how to use it, to improve multi-agent learning.

### 3.1 OFF-ENVIRONMENT RELABELLING

A useful perspective for determining how to relabel samples is to consider each multi-agent experience tuple separately, from the perspective of each agent, rather than as a single tuple received by

---

[1] which could also by described as a change to each agent's belief state.

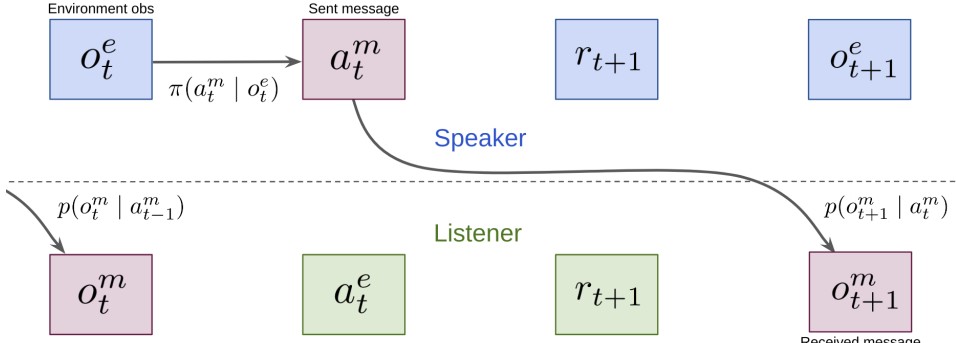

Figure 1: Consider a multi-agent experience tuple for a Listener agent receiving communication from a Speaker agent. In this simplified illustration the Speaker agent receives only environment observations, the Listener only receives communication. Our communication correction relabels the Listener's experience by generating a new message using the Speaker's current policy $\pi(a_t^m|o_t^e)$ and then generating the new Listener observation using the communication model $p(o_{t+1}^m|a_t^m)$. We shade in red the parts of the experience tuple which we relabel. Note that this relabelling only takes place for the Listener's sampled multi-agent experience, and not for the Speaker, as in this example the Speaker is not itself a Listener.

all agents as is commonly assumed. For a given agent's tuple, we can examine all the observed messages which constitutes its social environment (including even messages sent to other agents, which will be seen by a centralised critic). These were generated by past policies of other agents, and since then these policies may have changed due to learning or changes in exploration. Our first approach is therefore to relabel the communication tuple $(o_t^m, a_t^m, r_{t+1}^m, o_{t+1}^m)_i$ for agent $i$ by instead querying the current policies of other agents, replacing the communication actions accordingly and using the transition model to compute the new observed messages. For agent $i$ this procedure is:

$$\hat{a}_{\neg i,t}^m \sim \pi_{\neg i}(a_{\neg i,t}^m \mid o_{\neg i,t})$$
$$\hat{o}_{i,t+1}^m \sim p(o_{i,t+1}^m \mid \hat{a}_{\neg i,t}^m) \tag{5}$$

where $\neg i$ indicates agents other than $i$ and we use $\hat{z}$ to indicate that $z$ has been relabelled from its original value. Once the message has been relabelled for all agents, we can construct the overall relabelled joint observation by concatenation:

$$\hat{o}_{t+1} = o_{t+1}^e \oplus \hat{o}_{t+1}^m \tag{6}$$

We illustrate our Communication Correction (CC) idea in Figure 1 for the case of two agents, one sending out communication (the Speaker) and the other receiving communication (the Listener).

We experiment with feedforward policies which condition actions on the immediate observation, but this general idea could also be applied with recurrent networks using a history of observations $\hat{a}_{\neg i,t}^m \sim \pi_{\neg i}(a_{\neg i,t}^m \mid h_{\neg i,t})$; we discuss this in Future Work (Section 6). In our feedforward case, we sample an extra $o_{t-1}$ in order to determine (using the other agents' policies) the new $\hat{o}_t^m$, which allows us to relabel at the point of sampling from the replay buffer. Our relabelling approach could also straightforwardly be incorporated with attention-based models which also learn to *whom* to communicate (Das et al., 2019), but for our experiments we assume this is determined by the environment rather than the model.

### 3.2 ORDERED RELABELLING

One additional complexity to our approach is that the policies of the other agents may themselves be conditioned on received communication in addition to environmental observations. Our initial description ignores this effect, applying only a single correction. However, we can better account for this by sampling from the replay buffer an extra $k$ samples into the past. Starting from the

$k$'th sample into the past, we can set $\hat{o}_{t-k} = o_{t-k}$. Using Equations 5 and 6, we can then relabel according to:

$$\hat{o}_{t-k+1} \sim p(o_{t-k+1} \mid \hat{o}_{t-k}, o^e_{t-k+1}, \pi) \tag{7}$$

We iteratively apply this correction until $\hat{o}_{t+1}$ is generated. In general, this is an approximation if the starting joint observation $o_{t-k}$ depends on communication, but the corrected communication would likely be less off-environment than before. Furthermore, for episodic environments an exact correction could be found by correcting from the first time step of each episode.

In our experiments we consider a Directed Acyclic Graph (DAG) communication structure which also allows for exact corrections. In general a DAG structure may be expressed in terms of an adjacency matrix $D$ which is nilpotent; there exists some positive integer $n$ such that $D^m = 0, \forall m \geq n$. If $s$ is the smallest such $n$, we can set $k = s - 1$ which allows information to propagate from the root nodes of the DAG to the leaves. Agents which are not root nodes will not need $k$ updates for the influence of their messages to be propagated and so, for efficiency, for messages $c$ steps into the past we only generate messages which will have a downstream effect $c$ steps later (where $0 < c \leq k$). In general, we call our approach an Ordered Communication Correction (OCC), as opposed to our previous First-step Communication Correction (FCC) which only does one update.

### 3.3 Implementation

We include the full algorithm in Appendix A.9. We find in our experiments that relabelling can be done rapidly with little computational cost. Although different agents require different relabelling, the majority of the relabelling is shared (more so proportionally for increasing $N$). For simplicity, we can therefore relabel only a single multi-agent experience for the $N$ agents, and then correct for each agent by setting its own communication action back to its original value, as well as the downstream observations on the next time step. Once the sampled minibatch has been altered for each agent, we then use it for training an off-policy multi-agent algorithm. In our experiments we use MADDPG and Multi-Agent Actor-Critic (MAAC) (see Appendix A.4) (Iqbal & Sha, 2019) but our method could also be applied to other algorithms, such as value-decomposition networks (Sunehag et al., 2017) and QMIX (Rashid et al., 2018).

## 4 Results

We conduct experiments in the multi-agent particle environment[2] (MPE), a world with a continuous observation and discrete action space, along with some basic simulated physics. For common problems in the MPE, immediate observations summarise relevant history, such as velocity, such that optimal policies can be learned using feedforward networks, which we use for both policy and critic. We provide precise details on implementation and hyperparameters in Appendix A.1.

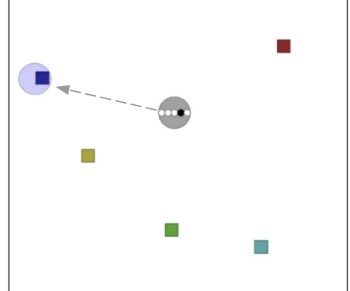

### 4.1 Cooperative Communication with 5 landmarks

Our first experiment, introduced by Lowe et al. (2017), is known as Cooperative Communication (Figure 2). It involves two cooperative agents, a Speaker and a Listener, placed in an environment with landmarks of differing colours. On each episode, the Listener must navigate to a randomly selected landmark; and both agents obtain reward proportional to its negative distance from this target. However, whilst the Listener observes its relative position from each of the differently coloured landmarks,

Figure 2: Cooperative Communication with 5 landmarks. Only the Speaker knows the target colour and must guide the Listener to the target landmark.

it does not know which landmark is the target. Instead, the colour of the target landmark is seen by an immobile Speaker. The Speaker sends a message to the Listener at every time step, and so successful performance on the task corresponds to it helping the Listener to reach the target.

---

[2]https://github.com/openai/multiagent-particle-envs

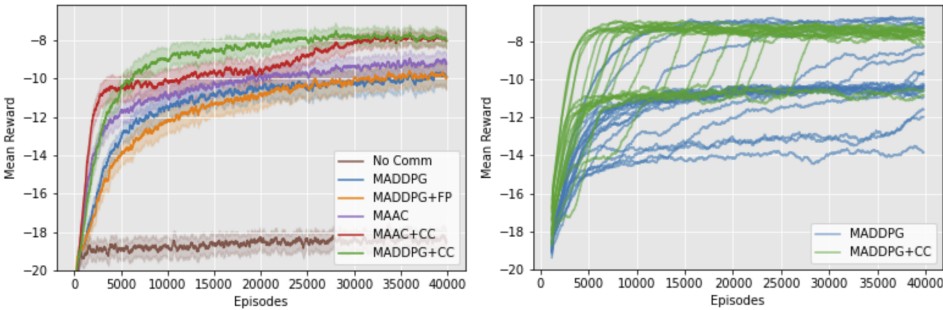

Figure 3: Cooperative Communication with 5 landmarks. (Left) MADDPG with communication correction (MADDPG+CC) substantially outperforms MADDPG (n=20, shaded region is standard error in the mean). (Right) Smoothed traces of individual MADDPG and MADDPG+CC runs. MADDPG+CC often has rapid improvements in its performance whereas MADDPG is slow to change.

Whilst Lowe et al. (2017) considered a problem involving only 3 landmarks (and showed that decentralised DDPG fails on this task), we increase this to 5. This is illustrated in Figure 2, which shows a particular episode where the dark blue Listener has correctly reached the dark blue square target, due to the helpful communication of the Speaker. We show performance on this task in Figure 3. Perhaps surprisingly both MADDPG and MAAC struggle to perfectly solve the problem in this case, with reward values approximately corresponding to only reliably reaching 4 of the 5 possible targets. We also implement a multi-agent fingerprint for MADDPG (MADDPG+FP) similar to the one introduced by Foerster et al. (2017), by including the training iteration index as input to the critic (see Appendix A.5), but we do not find it to improve performance. By contrast, introducing our communication correction substantially improves both methods, enabling all 5 targets to be reached more often. By looking at smoothed individual runs for MADDPG+CC we can see that it often induces distinctive rapid transitions from the 4 target plateau to the 5 target solution, whereas MADDPG does not. We hypothesise that these rapid transitions are due to our relabelling enabling the Listener to adapt quickly to changes in the Speaker's policy to exploit cases where it has learned to select a better communication action (before it unlearns this).

We also consider a version of Cooperative Communication in which randomly, 25% of the time, a sent message is not received. By relabelling with the same communication model, which also drops 25% of messages, we find again that MADDPG+CC does better than MADDPG (Appendix A.6).

## 4.2 HIERARCHICAL COMMUNICATION

We next consider a problem with a hierarchical communication structure, which we use to elucidate the differences between first-step and ordered communication corrections (MADDPG+FCC vs MADDPG+OCC). Our Hierarchical Communication problem (Figure 4) involves four agents. One agent is a Listener and must navigate to one of four coloured landmarks, but it cannot see what the target colour is. The remaining three Speaker agents can each see different colours which are certain not to be the target colour (indicated by their own colour in the diagram). However, only one Speaker can communicate with the Listener, with the rest forming a communication chain. To solve

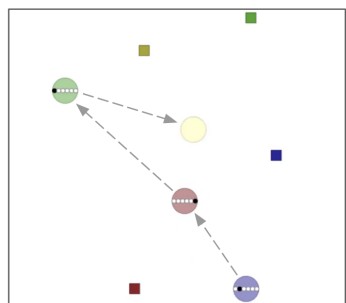

Figure 4: Hierarchical Communication. Three Speakers, with limited information and communicating in a chain, must guide the Listener to the target.

this task, the first Speaker must learn to communicate what colour it knows not to be correct, the middle Speaker must integrate its own knowledge to communicate the two colours which are not correct (for which there are 6 possibilities), and the final Speaker must use this to communicate the identity of the target landmark to the Listener, which must navigate to the target.

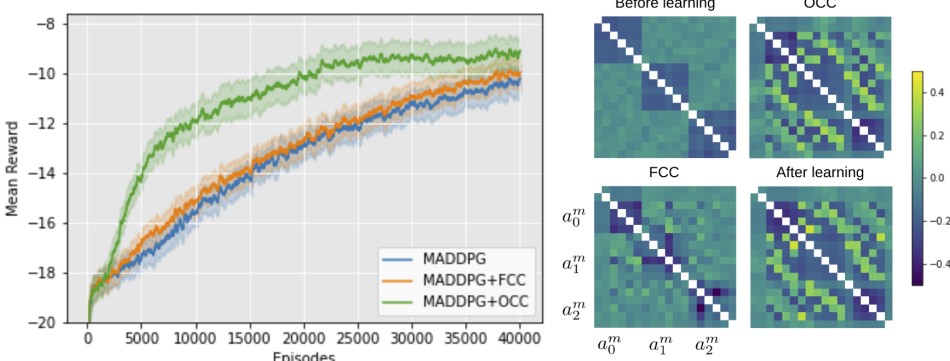

Figure 5: Hierarchical Communication. (Left) MADDPG+OCC substantially outperforms alternatives on this task (n=20). (Right) Correlation matrices for joint communication actions. Past communication from before learning is a poor reflection of communication after learning. OCC applied to past samples recovers this correlation structure whereas FCC only partially recovers this.

We analyse performance of MADDPG, MADDPG+FCC and MADDPG+OCC on this task in Figure 5. Whilst MADDPG+FCC applies the communication correction for each agent, it only does this over one time step, which prevents newly updated observations being used to compute the next correction. By contrast, the ordered MADDPG+OCC, with $k = 3$, starts from the root node, updates downstream observations and then uses the newly updated observations for the next update and so on (exploiting the DAG structure for more efficient updates). Our results show that MADDPG learns very slowly on this task and performs poorly, and MADDPG+FCC also performs poorly, with no evidence of a significant improvement over MADDPG. By contrast, MADDPG+OCC performs markedly better, learning at a much more rapid pace, and reaching a higher mean performance.

We would like to find out if the improved performance may be related to the ability of our method to recover correlations in communication. We therefore also examine the correlation matrices for the vector of joint communication actions. After having trained our MADDPG+OCC agents for $30,000$ episodes, we can compare samples of communication from the starting point of learning and after learning has taken place. We see that the correlation matrices are substantially different, with an intricate structure after learning reflecting the improved performance on the task. Without a communication correction, a 'before learning' sample would be unchanged, and the sample would therefore be a poor reflection of the current social environment. Using MADDPG+FCC we recover some of this structure, but there are still large differences, whereas MADDPG+OCC recovers this perfectly. This indicates that an ordered relabelling scheme is beneficial, and suggests that it may be increasingly important as the depth of the communication graph increases.

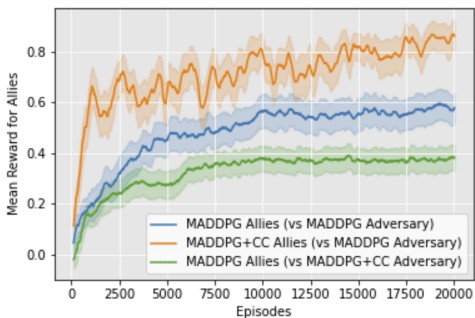

Figure 6: Covert Communication. When the Allies use the CC their performance is improved, whereas when their adversary uses it their performance is diminished (n=20).

### 4.3 COVERT COMMUNICATION

In our next experiment, we consider a competitive task first introduced by Lowe et al. (2017) called Covert Communication. In this task there are three agents; two Allies, one being a Speaker and another a Listener, and an Adversary. The Speaker sends messages which are received by both the Listener and the Adversary. However, whilst the Speaker would like the Listener to decode the message, it does not want the Adversary to decode the message. Both Speaker and Listener observe

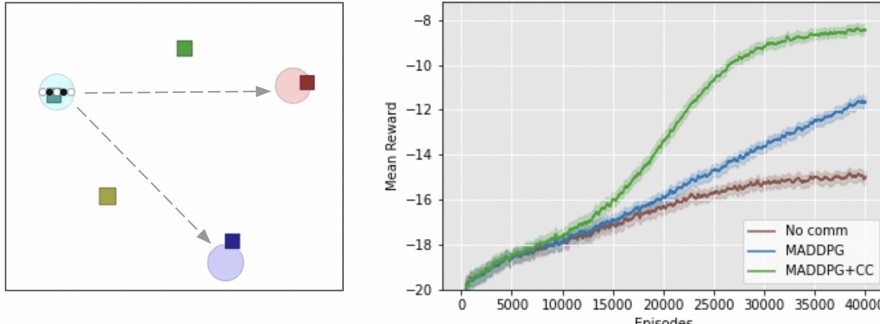

Figure 7: Cooperative Communication with Multiple Targets. (Left) One mobile Speaker agent and two Listener agents must each move to cover the target of their own colour. However, only the Speaker can see the target colours. For visualisation we overlay the Speaker's two discrete messages. (Right) The communication correction improves the performance of MADDPG (n=20).

a cryptographic 'key', which varies per episode and which the Adversary does not have access to. Reward for the Allies is the difference between how well the Listener decodes the message and how well the Adversary decodes the message (with 0 reward when both agents decode equally well).

One of the reasons this problem is interesting is because, unlike the previous Speaker-Listener problems which were ultimately cooperative, here there are competing agents. This is known to be able to induce large amounts of non-stationarity in communication policies and the environment. We therefore expect our experience relabelling to be effective in such situations, whether it be used for the Allies or the Adversary. Our results in Figure 6 demonstrate this; using the communication correction for the Allies but not the Adversary improves the Allies performance, whereas using it for the Adversary but not the Allies degrades the Allies performance. We find that this is because the communication correction allows agents to rapidly adapt their policies when their opponents change their policy to reduce their reward. For MADDPG+CC Allies, this involves ignoring the key and instead learning to exploit structure in the Adversary's behaviour to do better than if the opponent were random (see Appendix A.8 for an analysis).

## 4.4    COOPERATIVE COMMUNICATION WITH MULTIPLE TARGETS

In our final experiment, we apply our method to a situation in which the Speaker also takes environment actions and communicates cooperatively with two Listeners (Figure 7). There are 5 landmarks and each agent is assigned a randomly chosen target landmark. Each agent must navigate to cover its own target but only the Speaker sees the three target colours and so must communicate with the Listeners, sending different messages to each of them, to guide them to their targets. As with Cooperative Communication, we find that MADDPG+CC substantially outperforms MADDPG.

## 5    RELATED WORK

Multi-agent RL has a rich history (Busoniu et al., 2008). Communication is a key concept; however, much prior work on communication relied on pre-defined communication protocols. Learning communication was however explored by Kasai et al. (2008) in the tabular case, and has been shown to resolve difficulties of coordination which can be difficult for independent learners (Mataric, 1998; Panait & Luke, 2005). Recent work in the deep RL era has also investigated learning to communicate, including how it can be learned by backpropagating through the communication channel (Foerster et al., 2016; Sukhbaatar et al., 2016; Havrylov & Titov, 2017; Peng et al., 2017; Mordatch & Abbeel, 2018). Although we do not assume such convenient differentiability in our experiments, our method is in general applicable to this case, for algorithms which use a replay buffer. Other approaches which have been used to improve multi-agent communication include attention-based methods (Jiang & Lu, 2018; Iqbal & Sha, 2019; Das et al., 2019), intrinsic objectives (Jaques et al., 2019; Eccles et al., 2019) and structured graph-based communication (Agarwal et al., 2019).

Improvements to multi-agent experience replay were also considered by Foerster et al. (2017) who used decentralised training. Importance sampling as an off-environment correction was only found to provide slight improvements, perhaps due to the classical problem that importance ratios can have large or even unbounded variance (Robert & Casella, 2013), or with bias due to truncation. Here we focus specifically on communicated messages; this allows us to relabel rather than reweight samples and avoid issues of importance sampling. Of course, our method does not alter environment actions and so importance sampling for these may still be beneficial. In addition, it may in some cases be beneficial to condition our relabelled messages on these environment actions, perhaps using autoregressive policies (Vinyals et al., 2017).

Our approach also bears a resemblance to Hindsight Experience Replay (HER) (Andrychowicz et al., 2017), which can be used for environments which have many possible goals. It works by replacing the goal previously set for the agent with one which better matches the observed episode trajectory, which is helpful in sparse reward problems. This idea has been applied to hierarchical reinforcement learning (Levy et al., 2018), a field which can address single-agent problems by invoking a hierarchy of communicating agents (Dayan & Hinton, 1993; Vezhnevets et al., 2017). In such systems, goals are set by a learning agent, and one can also relabel its experience by replacing the goal it previously set with one which better reflects the (temporally-extended) observed transition (Nachum et al., 2018). Such ideas could naturally be combined with multi-agent HRL methods (Ahilan & Dayan, 2019; Vezhnevets et al., 2019), but they rely on communication corresponding to goals or reward functions. In contrast, our method can be applied more generally, to any communicated message.

## 6 FUTURE WORK

Similar to prior work using the MPE we experimented with problems which could be solved using feedforward policies (Lowe et al., 2017; Iqbal & Sha, 2019; Agarwal et al., 2019). In the future it would be interesting to extend our idea to richer problems solvable only by recurrent policies, which depend on the history of observations and actions. One strategy would be to build on existing methods which use experience replay with recurrent networks (Hausknecht & Stone, 2015; Kapturowski et al., 2018) whilst additionally relabelling when traversing the replay buffer. However, an additional challenge in this setting is that a communicating agent may signal its intended environmental actions prior to taking them, and a naive relabelling would not account for this. When relabelling in the recurrent setting it may therefore be beneficial to condition on these fixed future environment actions, or perhaps to consider the communication and environment as part of an hierarchical whole, and relabel them in a coordinated manner. These merit further investigation.

We also did not conduct experiments with loopy communication graphs. For problems of this nature, an exact correction requires relabelling from the start of each episode (although a $k$ steps back approximation may be sufficient in some cases). Whilst the general principal of the OCC updates are unchanged in this case, it may be more computationally efficient to sample and sequentially relabel full episodes of experience rather than individual experiences, as was done in this work.

## 7 CONCLUSIONS

We have shown how off-policy learning for communicating agents can be substantially improved by relabelling experiences. Our communication correction exploited the simple communication model which relates a sent message to a received message, and allowed us to relabel the received message with one more likely under the current policy. To address problems with agents who were both senders and receivers, we introduced an ordered relabelling scheme, and found overall that our method improved performance on both cooperative and competitive tasks. Our method provides a window into a general strategy for learning in non-stationary environments, in which acquired information in the present is used to alter past experience in order to improve future behaviour.

### ACKNOWLEDGMENTS

We would like to thank Danijar Hafner and Rylan Schaeffer for helpful comments. Sanjeevan Ahilan received funding from the Gatsby Computational Neuroscience Unit. Peter Dayan received funding from the Max Planck Society and the Alexander von Humboldt Foundation.

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

# A  APPENDIX

## A.1  HYPERPARAMETERS

For all algorithms and experiments, we used the Adam optimizer with a learning rate of $0.001$ and $\tau = 0.01$ for updating the target networks. The size of the replay buffer was $10^7$ and we updated the network parameters after every 100 samples added to the replay buffer. We used a batch size of 1024 episodes before making an update. We trained with 20 random seeds for all experiments and show using a shaded region the standard error in the mean.

For MADDPG we use the implementation of Iqbal & Sha (2019)[3]. For MADDPG and all MADDPG variants, hyperparameters were optimised using a line search centred on the experimental parameters used in Lowe et al. (2017) but with 64 neurons per layer in each feedforward network (each with two hidden layers). We found a value of $\gamma = 0.75$ worked best on Cooperative Communication with 6 landmarks evaluated after $50,000$ episodes. We use the Straight-Through Gumbel Softmax estimator with an inverse temperature parameter of 1 to generate discrete actions (see A.2).

For MAAC and MAAC+CC, we use the original implementation with unchanged parameters of Iqbal & Sha (2019)[4] . For our feedforward networks this corresponded to two hidden layers with 128 neurons per layer, 4 attend heads and $\gamma = 0.99$.

## A.2  DISCRETE OUTPUT WITH STRAIGHT-THROUGH GUMBEL SOFTMAX

We work with a discrete action space and so for MADDPG, following prior work (Lowe et al., 2017; Iqbal & Sha, 2019), we use the Gumbel Softmax Estimator (Jang et al., 2016; Maddison et al., 2016). In particular we use the Straight-Through Gumbel Estimator, which uses the Gumbel-Max trick (Gumbel, 1954) on the forward pass to generate a discrete sample from the categorical distribution, and the continuous approximation to it, the Gumbel Softmax, on the backward pass, to compute a biased estimate of the gradients.

The Gumbel Softmax enables us to compute gradients of a sample from the categorical distribution. Given a categorical distribution with class probabilites $\pi$, we can generate k-dimensional sample vectors $y \in \Delta^{n-1}$ on the simplex, where:

$$y_i = \frac{\exp((g_i + \log \pi_i)\beta)}{\sum_{j=1}^{k} \exp((g_j + \log \pi_j)\beta)} \tag{8}$$

for $i = 1, \ldots, k$, where $g_i$ are samples drawn from Gumbel(0,1)[5] and $\beta$ is an inverse temperature parameter. Lower values of $\beta$ will incur greater bias, but lower variance of the estimated gradient.

## A.3  MADDPG

Although the original MADDPG proposed the multi-agent policy gradient of Equation 1, this can cause over-generalisation when $\boldsymbol{a} \sim \mathcal{D}$ is far from the current policies of other agents (Wei et al., 2018). We use a correction introduced by Iqbal & Sha (2019) (for all variants of MADDPG), by sampling actions from the feedforward policy $\mu$ rather than the replay buffer, and providing this as input to the centralised critic.

$$\nabla_{\theta_i} J(\theta_i) = \mathbb{E}_{\boldsymbol{o} \sim \mathcal{D}, \boldsymbol{a} \sim \boldsymbol{\mu}}[\nabla_{\theta_i} \mu_i(o_i) \nabla_{a_i} Q_i^{\boldsymbol{\mu}}(\boldsymbol{o}, \boldsymbol{a})|_{a_i = \mu_i(o_i)}]. \tag{9}$$

One minor difference from Iqbal & Sha (2019) is that they generate the discrete action of other agents using the greedy action whereas we instead take a Gumbel-Softmax sample with $\beta = 1$.

---

[3]https://github.com/shariqiqbal2810/maddpg-pytorch
[4]https://github.com/shariqiqbal2810/MAAC
[5]using the procedure $u \sim \text{Uniform}(0, 1)$ and computing $g = -\log(-\log(u))$

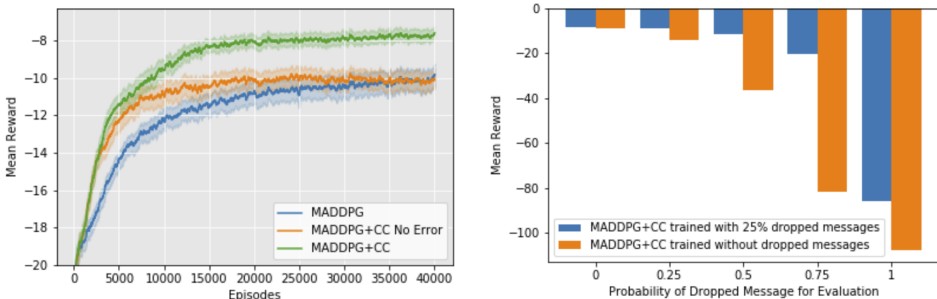

Figure 8: Cooperative Communication with Dropped Messages. (Left) Correcting communication whilst accounting for error improves the performance of MADDPG (n=20) (Right) We compare the performance of a model trained without dropped messages to one trained with 25% dropped messages. The model trained with dropped messages learns to be more robust to it than the model trained without (mean reward for each bar was averaged over 5000 episodes).

### A.4   MULTI-AGENT ACTOR ATTENTION CRITIC

Introduced by Iqbal & Sha (2019), MAAC learns the centralised critic for each agent by selectively paying attention to information from other agents. This allows each agent to query other agents for information about their observations and actions and incorporate that information into the estimate of its value function. In the case of MAAC+CC there is just a single shared critic and so we provide all altered samples to the same critic and do $N$ critic updates (where $N$ is the number of agents), and compare to MAAC which also does $N$ critic updates but with the unaltered samples.

### A.5   MULTI-AGENT FINGERPRINT

We incorporate a multi-agent fingerprint similar to the one introduced by Foerster et al. (2017). We do this by additionally storing the training iteration number. As training iteration numbers can get very large, and we find this hurts performance (we do not do use batch normalisation), we divide this number by $100,000$ before storing it (we typically train over $40,000$ episodes, which corresponds to $1,000,000$ iterations). We then train MADDPG in the conventional way, but additionally provide this number to the centralised critic. Foerster et al. (2017) also varied exploration and so provide the exploration parameter to the decentralised critics. We do not do this as we keep exploration fixed throughout training (with Gumbel $\beta = 1$).

### A.6   COOPERATIVE COMMUNICATION WITH DROPPED MESSAGES

We adapt the Cooperative Communication experiment such that 25% of sent messages are not received (the receiver sees all zeros). When correcting communication we compute what the new message would be and then randomly drop 25% of messages (MADDPG+CC). We drop messages by sampling from the communication model, irrespective of whether the original message that was sent was dropped or not (although an alternative approach of dropping the same messages could be explored in future). We find that applying this correction (MADDPG+CC) leads to better performance than MADDPG (Figure 8; left). We also test a relabelling approach which ignores the dropped message probability and simply relays them without error (MADDPG+CC No Error). As expected, this does much worse than MADDPG+CC, with performance approximately equal to MADDPG (although initial learning is faster).

Perhaps surprisingly, dropping messages 25% of the time does not provide an impediment to MADDPG+CC learning an effective policy. As the Listener can observe its own instantaneous velocity, it is possible that when no message is received it learns to continue moving towards the target it is already heading to, which works most of the time. Interestingly, if we compare a model trained with dropped messages to one trained without by evaluating it on the same task but with different probabilities of dropped messages, we find that the model trained with dropped messages is substantially more robust (Figure 8; right).

### A.7 EXPERIMENTAL ENVIRONMENTS

We provide a description of our environments in the main text but provide further information here. The episode length for all experiments was 25 time steps. For Cooperative Communication we alter the original problem by including two extra landmarks and by providing reward according to the negative distance from the target rather than the negative squared difference (as this elicits better performance in general). Our reward plots show the reward per episode.

For Hierarchical Communication, the information as to the target landmark is distributed amongst Speakers who cannot move and communicate in a chain to the Listener. As with Cooperative Communication we display rewards per episode and use the negative distance rather than the negative square distance.

For Covert Communication we plot the average reward per time step and use the original task, with one minor change of dividing the reward by 2. This ensures that the magnitude of the Allies rewards is less than or equal to 1 (for example, when the Allies are always right and the Adversary is always wrong the Allies reward is 1).

For Cooperative Communication with Multiple Targets, each agent must navigate to its target landmark but only the Speaker sees all three target colours. The Speaker sends separate discrete messages to each Listener. A shared reward equal to the summed negative distance of each agent from its target is provided to all agents. Our reward plots show the reward per episode divided by 3 (the number of agents).

### A.8 ANALYSIS OF COVERT COMMUNICATION

When used for the Allies, our communication correction substantially improved their performance. Interestingly, we found that rather than use the key, MADDPG+CC is able to learn communication actions which mislead its adversary into making the wrong choice, whilst simultaneously helping the Listener to make the right choice. This enables MADDPG+CC to achieve more reward than could be achieved against a random opponent (on average this would be 0.5). To emphasise this, we also trained agents on the same problem but without providing the Allies with the key. We find that when both Allies and the Adversary are trained using MADDPG the reward for the Allies is zero, because the Allies cannot adapt any faster than the Adversary. By contrast, performance for the MADDPG+CC Allies against a

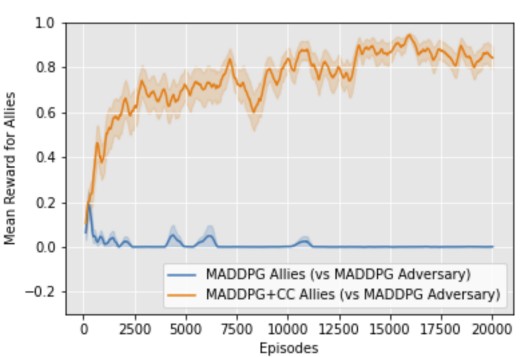

Figure 9: Covert Communication without the key (n=20). MADDPG+CC Allies outperform MADDPG Allies.

MADDPG opponent remains very good, with the MADDPG+CC Allies continuing to out-adapt the Adversary, causing it to perform even worse than random.

### A.9 ALGORITHM

We outline our Ordered Communication Correction algorithm here (First-step CC can be recovered by setting $K = 1$). We write it for the general case, although fewer updates can be used to achieve equivalent results if the communication graph is a DAG. For notational simplicity we use $\boldsymbol{a} \sim \boldsymbol{\pi}(\boldsymbol{a} \mid \boldsymbol{o})$ to indicate that each agent's policy $\pi_i$, receiving input $o_i$ was used to generate $a_i$ and thus collectively the joint action $\boldsymbol{a}$. Note that our description of the set of actions $\mathcal{A}_i = \mathcal{A}_i^e \times \mathcal{A}_i^m$ allows agents to both act and communicate simultaneously; in cases where agents only do one of these we allow for a 'no op' action.

An alternative way to define actions for each agent $\mathcal{A}_i$ is to divide them into disjoint environment actions $\mathcal{A}_i^e$ and explicit communication actions $\mathcal{A}_i^m$ such that $\mathcal{A}_i^e \cup \mathcal{A}_i^m = \mathcal{A}_i$ and $\mathcal{A}_i^e \cap \mathcal{A}_i^m = \emptyset$ (Lowe et al., 2019). To deal with this case one would require only minor alterations to the algorithm.

---

**Algorithm 1:** Ordered Communication Correction

---

**Given:**

- An off-policy multi-agent RL algorithm $\mathbb{A}$ using centralised training with decentralised policies
- One-step communication under model $p(\boldsymbol{o}^m_{t+1} \mid \boldsymbol{a}^m_t)$ which relates sent messages to received messages
- Feedforward policies

Initialise $\mathbb{A}$;
Initialise replay buffer $\mathcal{D}$;
Initialise number of steps to correct $K$;
**foreach** *episode* **do**
    Receive initial state $\boldsymbol{x}$ and joint observations $\boldsymbol{o}$;
    **foreach** *step of episode* **do**
        Select joint action $\boldsymbol{a} \sim \boldsymbol{\pi}(\boldsymbol{a} \mid \boldsymbol{o})$ using current policies (with exploration);
        Execute actions $\boldsymbol{a}$ which transitions environment state to $\boldsymbol{x}'$ and observe rewards $\boldsymbol{r}$, and next state $\boldsymbol{o}'$;
        Store $(\boldsymbol{o}, \boldsymbol{a}, \boldsymbol{r}, \boldsymbol{o}')$ in replay buffer $\mathcal{D}$;
        $\boldsymbol{x} \leftarrow \boldsymbol{x}'$;
        Sample from $\mathcal{D}$ at random indexes $\boldsymbol{t}$ a multi-agent minibatch $\boldsymbol{B}_t = (\boldsymbol{o}_{t-K}, \ldots, \boldsymbol{o}_t, \boldsymbol{a}_t, \boldsymbol{r}_{t+1}, \boldsymbol{o}_{t+1})$;
        $\hat{\boldsymbol{o}}_{t-K} = \boldsymbol{o}_{t-K}$;
        **for** $k = K, \ldots, 0$ **do**
            Compute new messages $\hat{\boldsymbol{a}}^m_{t-k} \sim \boldsymbol{\pi}(\boldsymbol{a}^m_{t-k} \mid \hat{\boldsymbol{o}}_{t-k})$;
            Compute new observed messages $\hat{\boldsymbol{o}}^m_{t-k+1} \sim p(\boldsymbol{o}^m_{t-k+1} \mid \hat{\boldsymbol{a}}^m_{t-k})$;
            $\hat{\boldsymbol{o}}_{t-k+1} = \boldsymbol{o}^e_{t-k+1} \oplus \hat{\boldsymbol{o}}^m_{t-k+1}$;
        **end for**
        $\hat{\boldsymbol{a}}_t = \boldsymbol{a}^e_t \oplus \hat{\boldsymbol{a}}^m_t$;
        **for** $i = 0, \ldots, N$ **do**
            Set agent $i$'s sent and received messages back to original value (where $\boldsymbol{l}$ is index of receiving agents);
            $\hat{\boldsymbol{a}}^{m,i}_t = \hat{\boldsymbol{a}}^m_{t,\neg i} \oplus \boldsymbol{a}^m_{t,i}$;
            $\hat{\boldsymbol{o}}^{m,i}_t = \hat{\boldsymbol{o}}^m_{t,\neg l} \oplus \boldsymbol{o}^m_{t,l}$;
            $\hat{\boldsymbol{o}}^{m,t}_{t+1} = \hat{\boldsymbol{o}}^m_{t+1,\neg l} \oplus \boldsymbol{o}^m_{t+1,l}$;
            $\hat{\boldsymbol{B}}^i_t = (\hat{\boldsymbol{o}}^i_t, \hat{\boldsymbol{a}}^i_t, \boldsymbol{r}_{t+1}, \hat{\boldsymbol{o}}^i_{t+1})$;
            Perform one step of optimisation for agent $i$ using $\mathbb{A}$ with relabelled minibatch $\hat{\boldsymbol{B}}^i_t$
        **end for**
    **end foreach**
**end foreach**

---

