# OpenReview forum: "Correcting experience replay for multi-agent communication"
_ICLR.cc/2021/Conference — ICLR 2021 Spotlight_

### Official Review · AnonReviewer4 · 2020-10-25
**Possibly a useful idea supported by interesting experiments, but the presentation lacks depth and detail**

**Rating:** 7
**Confidence:** 2

**Review:**

The paper considers a multi-agent reinforcement learning (MARL) scenario where agents take actions based on the current observation alone. The paper proposes a communication correction mechanism where, during the centralized training, messages there were received in the past from other agents are reevaluated according to the updated policy. This way old messages can be updated instead of discarded, which is more efficient overall.

The paper is clearly written and easy to follow. The experiments are enough to convince that the communication correction idea is effective. The idea is not very deep or insightful, so the significance of the contribution depends on how useful this trick will be in practice. This is completely fine, but since I'm not doing research in this area, it's hard for me to judge if this idea will be of value to other researchers, given the existing alternatives in the literature. Therefore my review focuses on the presentation of the idea.

I'm missing some intuition regarding why applying this correction is always "safe". Couldn't it somehow drift since it always gives so much importance to the new learned actions? what if for some period of time the new actions are worse than the old ones? Have you encountered any scenarios where this correction (at least if not properly tuned) degraded the performance?

Please elaborate on the assumption of actions that only depend on current observation. This seems very restrictive. How common is this assumption in the relevant literature? As a Markov strategy, it makes more sense if the state summarized the history of the game in some intelligent way. Is this the case for the scenarios you tested?

I think it can be insightful to also experiment with a scenario where communication is noisy. Specifically,  the scenario when there is a probability of p that a message is not received looks interesting to me. The reason is that this kind of randomness seems to be essentially different than the noiseless case, as opposed to a weak Gaussian noise. With this kind of noise, the sampling part of the algorithm will often sample a communication error when trying to relabel a message, effectively discarding the old message. Is this desirable or acceptable? Would one need to prevent this possibility if communication is random? Since questions like this arise (if it makes sense), I'd be careful not to make statements about how easily the method can be applied for more general scenarios (like with noise), especially since the paper is 100% empirical so we only see that what was tested works. Also, please define the ~ sign explicitly to avoid confusion.

Is this a realistic assumption that p(o|a) is known? it requires a model for the type of noise and communication failures that will occur in real-time, making the method not entirely model-free. Please discuss.

I didn't find Fig.1 very helpful. What does it contribute over (5)?

If you used the correction in Section A.3, why is (1) presented in the paper? seems a bit misleading to clarify that only in the appendix.

Please add a reference for "cheap talk".

The abstract should what is the basis for the claim that "it substantially improves...", so at least mention this is based on experiments.

"We analyze the performance of this task in Figure 3" - seems to be an overstatement, there is no analysis here, just reporting the results of the experiment.

---

> ### Author Response · Authors · 2020-11-18
> **Response to Reviewer 4 (Part I)**
>
> Thank you very much for your helpful review. Here are our answers to your questions:
>
> >I'm missing some intuition regarding why applying this correction is always "safe". Couldn't it somehow drift since it always gives so much importance to the new learned actions? what if for some period of time the new actions are worse than the old ones? Have you encountered any scenarios where this correction (at least if not properly tuned) degraded the performance?
>
>
> A focus of this work is to address issues of non-stationarity of the environment in multi-agent learning, caused by the changing policies of other agents. Updating an agent’s policy using data from an old environment would be ‘dangerous’ because that environment may be very different from the current environment. Our method alters past messages to make old experience look more similar to samples from the current environment, which helps learning.
>
> It is true to say that learning even with the correction could result in new actions being worse than old ones, although so far we have not found this in our experiments. However, this issue is not particular to our method - in general, training in deep RL can be unstable, even in the single-agent case, and so declining performance remains a possibility.
>
> > Please elaborate on the assumption of actions that only depend on current observation. This seems very restrictive. How common is this assumption in the relevant literature?
>
> The use of only feedforward policies is common in the literature, and frequently used for published work using the multi-agent particle environment (Lowe et al., 2017; Iqbal & Sha, 2019; Agarwal et al., 2019). However, this does deserve elaboration and we now include a Future Work section which discusses extending our idea to recurrent networks.
>
> > As a Markov strategy, it makes more sense if the state summarized the history of the game in some intelligent way. Is this the case for the scenarios you tested?
>
> Exactly as you say, the scenarios we test allow for an optimal policy to be learned which conditions only on immediate observations, because the state is summarising the history of the game in a useful way. For example, agents which move in our experiments get to immediately perceive their instantaneous velocity.
>
> >I think it can be insightful to also experiment with a scenario where communication is noisy. Specifically, the scenario when there is a probability of p that a message is not received looks interesting to me. The reason is that this kind of randomness seems to be essentially different than the noiseless case, as opposed to a weak Gaussian noise. With this kind of noise, the sampling part of the algorithm will often sample a communication error when trying to relabel a message, effectively discarding the old message. Is this desirable or acceptable? Would one need to prevent this possibility if communication is random? Since questions like this arise (if it makes sense), I'd be careful not to make statements about how easily the method can be applied for more general scenarios (like with noise), especially since the paper is 100% empirical so we only see that what was tested works. Also, please define the ~ sign explicitly to avoid confusion.
>
> This is an interesting idea for an experiment and we have now run this for Cooperative Communication where 25% of sent messages are not received (see Appendix A.6, Cooperative Communication with Dropped Messages, also referenced in Section 4.1). Our expectation was that our method should handle this case and the results demonstrate this (see MADDPG+CC vs MADDPG). Relabelling whilst also dropping messages is desirable as we want the receiving agent to be able to act effectively in that situation (but we also mention how this could benefit from further investigation). We also test relabelling without dropping messages (violating the communication model) and find that performance is worse (see MADDPG+CC No Error), in line with our expectations. We also now define the ~ sign as requested.
>
> > Is this a realistic assumption that p(o|a) is known? it requires a model for the type of noise and communication failures that will occur in real-time, making the method not entirely model-free. Please discuss.
>
> We think that in many situations this will be a realistic assumption, both in simulation and in real life. Modern communication technologies are advanced, and usually allow messages to be sent robustly (using error correction) and predictably. In cases where communication fails, this can often be easily verified. We therefore believe our assumption is a reasonable one to make and will allow for application in a wide variety of problems. As you rightly point out, this adds a model-based component to our work, and we believe that leveraging this simple model effectively is one of the reasons we see improved performance.

---

> > ### Comment · AnonReviewer4 · 2020-11-20
> > **The new experiment strengthens the contribution**
> >
> > Thank you for your detailed and meticulous response and especially for the new interesting experiment. I am also happy to see that the other reviewers find the contribution significant and useful for researchers in the field. Therefore, I'm raising my score to 7 since  I believe that the paper should be accepted.
> >
> > Some responses to the major issues:
> >
> > Markov Strategies: I agree, I just wonder whether it is worth mentioning that the state isn't a given physical truth and can be designed to compensate for this limited family of strategies. After all, this degree of freedom is implicit in the mathematical definition. Giving an example of how the state summarizes the relevant information well in your experiments could also be nice. However, if it is common practice for MARL then I understand why it is not necessary to get into that.
> >
> > Communication Failures: I think that the new experiment is very encouraging. It was not obvious to me that the results would be positive, so I think that this new contribution makes the paper stronger and demonstrates that the proposed trick more robust than one might suspect.
> >
> > Known p(o|a): Overall, I agree that this is a reasonable assumption, but it is also a strong one. As you nicely phrased it, it might be a key reason why the new trick leads to improved performance. Even if modern communication is well established, it uses models that depend on the environment (obstacles, the velocity of objects, external interference from other systems). As such, assuming that p(o|a) is known is like assuming that something about the environment is known. I think that discussing that in the paper can be both fair and an interesting challenge for researchers to relax this assumption.

---

> > > ### Author Response · Authors · 2020-11-22
> > > **Response to Reviewer 4's comment**
> > >
> > > Thank you for your comment and for raising your score.
> > >
> > > >Markov Strategies: I agree, I just wonder whether it is worth mentioning that the state isn't a given physical truth and can be designed to compensate for this limited family of strategies. After all, this degree of freedom is implicit in the mathematical definition. Giving an example of how the state summarizes the relevant information well in your experiments could also be nice. However, if it is common practice for MARL then I understand why it is not necessary to get into that.
> > >
> > > Yes, we agree that that could be clearer. At the start of the Results section (Section 4) we have now added the extra sentence: "For common problems in the MPE, immediate observations summarise relevant history, such as velocity, such that optimal policies can be learned using feedforward networks, which we use for both policy and critic."
> > >
> > > > Communication Failures: I think that the new experiment is very encouraging. It was not obvious to me that the results would be positive, so I think that this new contribution makes the paper stronger and demonstrates that the proposed trick more robust than one might suspect.
> > >
> > > Yes, thank you, it was certainly an interesting experiment to consider. To help better understand the Cooperative Communication with Dropped Messages experiment we have now included one extra plot in Figure 8 in the Appendix, along with some small changes to the text. The new plot compares a MADDPG+CC model trained with 25% Dropped Messages to a MADDPG+CC model trained without Dropped Messages by evaluating them on the same task but with different probabilities of dropping messages. The plot shows that the model trained with dropped messages is more robust.
> > >
> > > Although this is not a key result, we hope it sheds some extra light on the new experiment in the Appendix.
> > >
> > > >Known p(o|a): Overall, I agree that this is a reasonable assumption, but it is also a strong one. As you nicely phrased it, it might be a key reason why the new trick leads to improved performance. Even if modern communication is well established, it uses models that depend on the environment (obstacles, the velocity of objects, external interference from other systems). As such, assuming that p(o|a) is known is like assuming that something about the environment is known. I think that discussing that in the paper can be both fair and an interesting challenge for researchers to relax this assumption.
> > >
> > > Agreed. Although due to space constraints we have not been able to discuss these ideas in depth, we hope we have touched on them sufficiently for our readers to consider when communication models can or cannot be easily applied.

---

> ### Author Response · Authors · 2020-11-18
> **Response to Reviewer 4 (Part II)**
>
> >I didn't find Fig.1 very helpful. What does it contribute over (5)?
>
> We wanted to illustrate the key idea for a sufficiently simple example so that updates could be seen visually (without too many arrows). The diagram also highlights that we relabel both o_t^m and o_t^{m+1} using Equation 5, which is not obvious from looking at Equation 5 alone.
>
> > If you used the correction in Section A.3, why is (1) presented in the paper? seems a bit misleading to clarify that only in the appendix.
>
> As it was the Background section, we wanted to present the original MADDPG algorithm, but we agree it could be more clear. To improve the clarity, we now write at the end of this section: ‘we use this algorithm with some additional changes (see Appendix A.3 for details)'.
>
> > Please add a reference for "cheap talk".
>
> We now reference the 1996 paper on cheap talk by Farrell and Rabin.
>
> > The abstract should what is the basis for the claim that "it substantially improves...", so at least mention this is based on experiments.
>
> We have changed the abstract to say: “We find in our experiments that it substantially improves the ability of communicating MARL systems to learn across a variety of cooperative and competitive tasks”
>
> > "We analyze the performance of this task in Figure 3" - seems to be an overstatement, there is no analysis here, just reporting the results of the experiment.
>
> We replace the word ‘analyse’ with the word ‘show’.

---

### Official Review · AnonReviewer2 · 2020-10-27
**Interesting method, could use more discussion of when it applies.**

**Rating:** 7
**Confidence:** 4

**Review:**

---- Summary ----
The paper proposes a method for modifying an experience replay when learning in communication environments, by relabelling messages using the latest policy.

---- Reasons for score ----
The paper addresses the problem of non-stationarity in an important class of multiagent environment. The correction proposed is simple, and effective in the domains it is tested in. My main concern is how broadly the method applies, which I am uncertain of from the paper.

---- Pros ----
The paper provides a simple way to better leverage replay data for communication environments, addressing non-stationarity in those environments. This is an important problem in multiagent environments.

The experiments show improvements in learning speed and final reward in some communication domains. These cover a few important cases for the algorithm, including hierarchical communication and communication with an adversarial listener.

The paper is well situated in the literature on emergent communication, and it is clear and well written throughout.

---- Cons ----
My main worry is that the paper leaves me uncertain on when the method can be applied. The OCC algorithm suggests that a necessary condition is that the message graph is acyclic. However, this limitation is not explicitly discussed, and I am also unsure whether an acyclic message graph is a sufficient condition. For example, does the method apply with multiple listeners acting in the same environment, or when the speaker also acts? It would strengthen the paper to be more precise about the settings where these algorithms can be applied and can be expected to help.

The discussion on Covert Communication in the appendix significantly changed how I understood the results of this experiment; in particular, the agents have not solved the task intended in the environment (using the key to communicate), but instead appear to be constantly changing strategy to outpace the listener - with the key being irrelevant. This is hinted at in the main text, but I think it is central enough to the interpretation of the experiment that it should be moved there.

---

> ### Author Response · Authors · 2020-11-20
> **Response to Reviewer 2**
>
> Thank you very much for your helpful review.
>
> >My main worry is that the paper leaves me uncertain on when the method can be applied. The OCC algorithm suggests that a necessary condition is that the message graph is acyclic. However, this limitation is not explicitly discussed, and I am also unsure whether an acyclic message graph is a sufficient condition. For example, does the method apply with multiple listeners acting in the same environment, or when the speaker also acts? It would strengthen the paper to be more precise about the settings where these algorithms can be applied and can be expected to help.
>
> We have taken your concerns on board regarding being more precise about when the method can be applied. To answer your question about environments with multiple listeners and the speaker also acting, we introduce a new experiment called Cooperative Communication with Multiple Targets which has these properties, and find that the communication correction substantially improves performance. We explain the details in the new Section 4.4 and show the results in the new Figure 7. We then outline in a new Future Work section the kinds of problems we do not have results for in this paper and leave to future work.
>
> The answer to your specific question regarding the OCC can be found in Future Work. Although we applied our method to DAGs, we expect it to work for loopy graphs if relabelling starts from the beginning of each episode (although a k steps back approximation may be sufficient in some cases). This is because the initial observation at the beginning of an episode will not depend on previously sent messages, whereas this may not be true for later observations. We also point out that for loopy graphs it may be more computationally efficient to relabel full episodes of experiences rather than individual experiences, as we have done in this work.
>
> > The discussion on Covert Communication in the appendix significantly changed how I understood the results of this experiment; in particular, the agents have not solved the task intended in the environment (using the key to communicate), but instead appear to be constantly changing strategy to outpace the listener - with the key being irrelevant. This is hinted at in the main text, but I think it is central enough to the interpretation of the experiment that it should be moved there.
>
> We have now moved this to the main text as you suggest, to make the interpretation of the experiment clearer. We add to the main text: “For MADDPG+CC Allies, this involves ignoring the key and instead learning to exploit structure in the Adversary’s behaviour to do better than if the opponent were random (see Appendix A.8 for an analysis).”

---

> > ### Comment · AnonReviewer2 · 2020-11-20
> > **Comment on authors response**
> >
> > Thank you, these updates address my concerns well.
> >
> > The new experiment is helpful; though the lack of interaction between the agents' environment actions means it does not fully address the sort of problem I was thinking of - Reviewer 1's example domain is an excellent example of this. However, the new Future Work section discusses these limitations.
> >
> > The updated text on Covert Communication addresses my concerns with the presentation of that result.

---

> > > ### Author Response · Authors · 2020-11-20
> > > **Comment on Reviewer 2's response**
> > >
> > > Thank you, we are glad that these updates have addressed your concerns well.

---

### Official Review · AnonReviewer1 · 2020-10-27
**Accept: Correcting experience replay for multi-agent communication**

**Rating:** 8
**Confidence:** 3

**Review:**

Summary:

This paper considers communication games when agents use experience replay. The agents' communication protocol may change over time, leaving outdated symbols in the replay buffer which are then trained on. This paper proposes replacing the old communication actions with up-to-date actions as the transitions are sampled, and shows that this leads to greatly improved convergence speed and higher performance plateaus.

--------------------

Positives:

- The problem of multiagent communication and how to learn it is important and relevant to the ICLR community. The solution presented here seems like a natural fit with the problem and popular agent architectures and is well presented.

- The paper is well motivated and well written.  Overall it was an enjoyable and easy read!

- The experiments in Figures 4, 5, and 6 seem like great choices to show the strengths of the approach.  They're simple, well described, and well targeted.

--------------------

Negatives:

- I feel like there's a pretty obvious question about "What happens in richer domains?" that (unless I missed it) isn't addressed in the paper - I'll expand on that in my 'Questions to clarify recommendation' section below. While the technique seems to work very well in the experiments chosen for the paper, I wish the paper touched a bit more on upcoming challenges, possible foreseen problems, and next steps.

--------------------

Recommendation and Justification:

Overall, I feel like this was a strong paper and should be accepted. My only real negative was that I am excited to know more about what comes next.

--------------------

Questions to clarify recommendation:

The three environments presented in the paper, if I've understood them correctly, are pretty straightforward in that 1) the speaker only has communication actions (and no environment actions), and 2) seems to only have one consistent message to communicate during the entire episode (after perhaps waiting to receive a message from others, in Hierarchical Communication). But right from the abstract onwards, I was wondering about possible problems in richer domains, where it seems like this technique could be harmful. Specifically, what if by updating the old communication action to one chosen by the current policy, we present a communication action that no longer aligns with the old environment action, which we do not update?  I felt like this was a pretty natural question, but unless I missed it, the paper doesn't mention possible problems like this.

I'll ground this in an example, similar to Cooperative Communication.  Imagine a two-player gridworld where the players cannot see each other, but are rewarded for arriving at the same map location.  Similar to Bach and Stravinsky / Battle of the Sexes, each player has a different preference over locations, but being at the same location is most important. Let's call the locations Left and Right.  To enable coordination, let one player be a Speaker that can take communication actions to signal the other player as to where they should meet in that episode. While that permits greedy Speaker policies (always announce their preferred location and then go there) and greedy Listener policies (always go to their preferred location, regardless of Speaker's announcement), it would also allow the speaker to arrange a correlated equilibrium: announce a randomly chosen location in each episode and then go there, to maximize joint reward beyond any greedy Nash equilibrium policy.

Here's where I see a possible failure with the technique in this paper.  Assume that the replay buffer contains an episode where the speaker emitted symbol L (for left) and then took environment actions to move to the Left location.  Later in training, using the technique presented here, we might sample this experience, update the symbol to L', and still move Left.  As described in this paper, I would expect that should work, and converge faster than by using the out-of-date symbol L.  However, it seems possible that the newer Speaker policy might prefer to move Right on that episode instead.  Updating the symbol would change it from old L to new R', but since the technique does not (and cannot, without a world model) update the environment actions, it seems like the listener and speaker would then train on this misaligned tuple of communication action R' and environment actions to move Left.  I would expect this confusing example to be much worse than training on the original example with the outdated but still aligned communication actions.

More generally: how can we make sure that the updated communication actions still align in intent with the agent's environment actions that we cannot change? It seems like conditioning the communication action on the agent's environment action for that timestep might help, but would only be a partial solution: if an agent must speak now but take their first significant environment action in the future, we would have the same problem.

My questions regarding this point are:
- Do you agree that this could be a problem in richer environments than those presented in the paper?
- If so, do you foresee an easy solution, or will this be a challenge for future work?

I think the paper is strong enough as-is, and does not need an experiment in this paper to investigate richer games like this.  However, if the authors agree that the technique could fail to help or could harm convergence in richer settings than those presented in the paper to support the technique, then I think a couple of sentences about future challenges and future work are warranted.

-------------------

Issues and Suggestions:

- Nit: Pg2, Experience Replay. The first sentence describes the agent as receiving (s_t, a_t, ..., s_t+1) at each time step. Should this be (o_t, ..., o_t+1), since the agent receives observations and not environment states?

- Typo: Pg2, MADDPG.  'uses deterministic polices' --> policies

- Suggestion: Pg3, Methods section and Equation 4. Equation 4 describes the tuple as containing r^e_t+1, r^m_t+1, but the text in the paragraph above only mentions r_t'+1, and the text below only indirectly clarifies what r^e and r^m are when it describes the cheap talk setting where r^m=0.  This threw me for a while when I read the equation, and scanned back up the page to try to see where r^e and r^m were defined, and they aren't. I suggest changing the sentence in the previous paragraph from "receives rewards r_t'+1" to something like "receives rewards r_t'+1 (split into an environmental reward r^e and a messaging cost r^m)..." to clarify this before the symbols are used.

- Typo: Pg4, Ordered Relabelling. "may themselves by conditioned" --> "may themselves be conditioned"

- Clarify: Pg4, under equation 6. The sentence "...we sample an extra o^m_t-1 in order to determine (using the other agents' policies) the new \^{o}^m_t, which allows us to relabel...". I don't understand what this sentence is trying to say. Which player is this for? The symbol \^{o}^m_t doesn't appear in equations 5 or 6, so I don't understand what sampling an extra o^m_t-1 would do, since it's to compute a symbol that doesn't connect with the equations being discussed. Maybe I'm just missing something obvious, but I spent a couple of minutes trying to figure this out, before giving up and moving on.

- Nit: Pg5, Implementation. Extremely minor, but the phrasing "we can therefore only relabel..." suggests a limitation of the approach (e.g., we are only able to do this...) whereas I think you're suggesting a performance win (we can do this using only...). I feel like flipping the words to "we can therefore relabel only a single..." better communicates that.

- Typo: Pg9 and 10, References. In both of the references including Pieter Abbeel, his affiliation is prefixed (OpenAI Pieter Abbeel). No other authors' affiliations are listed, so this just seems like a .bib typo.

---

> ### Author Response · Authors · 2020-11-20
> **Response to Reviewer 1 (Part I)**
>
> Thank you very much for your helpful review. We hope to address the issues you have raised here:
>
> >I feel like there's a pretty obvious question about "What happens in richer domains?" that (unless I missed it) isn't addressed in the paper - I'll expand on that in my 'Questions to clarify recommendation' section below. While the technique seems to work very well in the experiments chosen for the paper, I wish the paper touched a bit more on upcoming challenges, possible foreseen problems, and next steps.
>
> >Do you agree that this could be a problem in richer environments than those presented in the paper?
>
> We agree that you have raised an important point and we have given this more attention in our revised version, primarily through the inclusion of a Future Work section. The Bach and Stravinsky example summarises perfectly your concern regarding richer domains, and we appreciate the thought you put into it. We had in fact considered a similar kind of problem (a thought experiment involving colliding cars), prior to submission, and alluded to it by writing: "In addition, it may in some cases be beneficial to condition our relabelled messages on these environment actions, perhaps using autoregressive policies (Vinyals et al., 2017)".
>
> However, this was from the perspective of feedforward networks, whereas your example references memory-based agents for which these issues may become more salient. In particular, in the future we would like to develop a recurrent relabelling scheme which is effective on problems where an agent learns to signal its intended environmental actions prior to taking them.
>
> >More generally: how can we make sure that the updated communication actions still align in intent with the agent's environment actions that we cannot change? It seems like conditioning the communication action on the agent's environment action for that timestep might help, but would only be a partial solution: if an agent must speak now but take their first significant environment action in the future, we would have the same problem.
>
>  We make suggestions for potential strategies for recurrent (memory-based) problems in Future Work. These include conditioning relabelling on fixed future environment actions or considering  the communication and environment as part of an hierarchical whole.
>
> > If so, do you foresee an easy solution, or will this be a challenge for future work?
>
> Whilst a solution may not be straightforward, we do think it is achievable, and given the success of our approach in the feedforward case we believe this would be exciting to explore in Future Work. A sketch of a potential solution could perhaps be:
>
> 1. Using the recurrent net, which conditions on fixed environment actions in the past, replace the sequences of messages in the sampled episode
> 2. Simultaneously, compute and store the joint likelihood of the sequence of actions taken by the agent (including environment actions and messages sent).
> 3. Repeat c times, where c is a hyperparameter which determines the trade-off between computation cost and relabelling which better accounts for future environment actions
> 4. Choose the relabelling which maximises the joint likelihood of relabelled messages and environment actions
>
> A solution of this kind would of course need substantial further experimentation to test it, which is beyond the scope of this paper. We nevertheless hope that our idea of relabelling as an off-environment correction encourages future research in this direction.
>
> > I think the paper is strong enough as-is, and does not need an experiment in this paper to investigate richer games like this. However, if the authors agree that the technique could fail to help or could harm convergence in richer settings than those presented in the paper to support the technique, then I think a couple of sentences about future challenges and future work are warranted.
>
> Thank you. We hope the new Future Work section addresses your concerns.

---

> ### Author Response · Authors · 2020-11-20
> **Response to Reviewer 1 (Part II)**
>
> Thank you for highlighting various small issues. We go through them here:
>
> > Nit: Pg2, Experience Replay. The first sentence describes the agent as receiving (s_t, a_t, ..., s_t+1) at each time step. Should this be (o_t, ..., o_t+1), since the agent receives observations and not environment states?
>
> We have made the change you suggest.
>
> > Typo: Pg2, MADDPG. 'uses deterministic polices' --> policies
>
> Thanks, changed.
>
> > Suggestion: Pg3, Methods section and Equation 4. Equation 4 describes the tuple as containing r^e_t+1, r^m_t+1, but the text in the paragraph above only mentions r_t'+1, and the text below only indirectly clarifies what r^e and r^m are when it describes the cheap talk setting where r^m=0. This threw me for a while when I read the equation, and scanned back up the page to try to see where r^e and r^m were defined, and they aren't. I suggest changing the sentence in the previous paragraph from "receives rewards r_t'+1" to something like "receives rewards r_t'+1 (split into an environmental reward r^e and a messaging cost r^m)..." to clarify this before the symbols are used.
>
> We have made the change you suggest, to improve clarity.
>
> > Typo: Pg4, Ordered Relabelling. "may themselves by conditioned" --> "may themselves be conditioned"
>
> Thanks, changed.
>
> > Clarify: Pg4, under equation 6. The sentence "...we sample an extra o^m_t-1 in order to determine (using the other agents' policies) the new ^{o}^m_t, which allows us to relabel...". I don't understand what this sentence is trying to say. Which player is this for? The symbol ^{o}^m_t doesn't appear in equations 5 or 6, so I don't understand what sampling an extra o^m_t-1 would do, since it's to compute a symbol that doesn't connect with the equations being discussed. Maybe I'm just missing something obvious, but I spent a couple of minutes trying to figure this out, before giving up and moving on.
>
> Thanks for spotting this typing error, we should have said "In our feed-forward case, we sample an extra \bm{o}\_{t-1} in order to determine (using the other agents' policies) the new \bm{\hat{o}}^m\_{t}". The $m$ subscript should not have been there, and we should have added the \hat.
>
> >Nit: Pg5, Implementation. Extremely minor, but the phrasing "we can therefore only relabel..." suggests a limitation of the approach (e.g., we are only able to do this...) whereas I think you're suggesting a performance win (we can do this using only...). I feel like flipping the words to "we can therefore relabel only a single..." better communicates that.
>
> We have made the suggested change.
>
> > Typo: Pg9 and 10, References. In both of the references including Pieter Abbeel, his affiliation is prefixed (OpenAI Pieter Abbeel). No other authors' affiliations are listed, so this just seems like a .bib typo.
>
> Good spot, we have made the change to the .bib.

---

### Official Review · AnonReviewer3 · 2020-10-29
**Intriguing and effective novel approach to off-policy learning**

**Rating:** 8
**Confidence:** 3

**Review:**

The authors present a fun and effective idea to translate a peer's message in terms of one agent's own experience. The benefit of doing so makes sense intuitively and is verified to be effective empirically.


Strengths:

+ The motivation is clear, and the key idea is well-presented. Paper is positioned well in relevant works of communication-aided MARL research.

+ Evaluation is thorough and indicative of the authors' claims.


Major Concerns:

- The reviewer has yet to discover a major issue with the paper with regard to its correctness, contribution, novelty, and effectiveness.

---

> ### Author Response · Authors · 2020-11-20
> **Response to Reviewer 3**
>
> Thank you very much for your review, we are glad you enjoyed the paper.

---

### Author Response · Authors · 2020-11-18
**We add two new experiments and a Future Work section in our revised version**

We thank the reviewers for their insightful and constructive comments. To address their concerns we have introduced two extra experiments and a Future Work section. Our first new experiment (Section 4.4 and Figure 7) is called ‘Cooperative Communication with Multiple Targets’. AnonReviewer2 asked if our method could be applied in cases where the Speaker itself takes environment actions and where there are multiple Listeners. In this experiment the Speaker must move to cover its own target but also communicate to two Listeners to help them cover their targets as well. We show that MADDPG+CC outperforms MADDPG on this experiment, demonstrating that our method can work for these kinds of problems.

Our second experiment, which we include in Appendix A.6 and reference in Section 4.1 is called ‘Cooperative Communication with Dropped Messages’. AnonReviewer4 asked if our method would work if sent messages were not reliably received. In this experiment we randomly cause 25% of messages not to be received, and use this aspect of the communication model for relabelling. We find strong performance improvements for MADDPG+CC which relabels whilst also randomly dropping 25% of messages. We also show that relabelling without dropping messages nullifies those improvements.

In our Future Work section (Section 6) we seek to highlight problems we have not addressed in this work and would benefit from future research. AnonReviewer1 presented a thoughtful example of when relabelling naively may be misleading (we had also considered a related problem when referencing auto-regressive policies in Related Work). AnonReviewer1’s example involved an agent communicating its intentions and later selecting its environmental actions based on those intentions. This is a problem which becomes salient in the case where agents use recurrent policies, and in the future we would like to develop methods which extend our idea to this case. We note this in the Future Work section, and provide an indication of approaches which may be taken to resolve this in the future. We also note that other recent multi-agent work have used only feed-forward networks in their experiments (Lowe et al., 2017; Iqbal & Sha, 2019; Agarwal et al., 2019).

We also discuss in Future Work the possibility of applying our method to loopy communication graphs, as requested by AnonReviewer2. We point out that whilst such experiments are left for future work, we could use the OCC to update from the start of each episode, and that we would expect our method to work for these problems.

Overall, we hope these corrections address the important points raised by the reviewers. Other small points and corrections were also raised and we have also addressed these in the revised version of the paper (we will make note of these in our response to each reviewer).

EDIT: We have made an additional small change to the paper by adding a new plot and description for Figure 8 in the Appendix, to help better understand the Cooperative Communication with Dropped Messages experiment. This plot compares a MADDPG+CC model trained with 25% Dropped Messages to a MADDPG+CC model trained without Dropped Messages by evaluating them on the same task but with different probabilities of dropping messages. The plot shows that the model trained with dropped messages is more robust. Although this is not a key result, we hope it sheds some extra light on the new experiment in the Appendix.

---

### Decision · Program_Chairs · 2021-01-07
**Final Decision**

**Decision:**

Accept (Spotlight)

**Comment:**

This work proposes a simple and intuitive way to improve how to learn a communication protocol off-policy in the non-stationary situation in which messages received in the past do not reflect an agent's current policy. The authors introduce a communication correction that relabels the received message adjusting it to the current policy. The authors show that this method, besides being simple, is effective in a number of experiments. As observed by some reviewers, an issue with the method is that it is not clear how it would scale up to more complex environments than those considered. However, the authors addressed the concerns during the response phase, both adding new experiments, and with a clear statement of what are the outstanding issues. The paper is certainly a clever and solid contribution to the area of multi-agent communication learning, and I am strongly in favour of accepting it.